



# A model-data assessment of the role of Southern Ocean processes in the last glacial termination

Roland Eichinger[1], Gary Shaffer[2,3,4], Nelson Albarrán[5], Maisa Rojas[1], and Fabrice Lambert[6]

[1]Department of Geophysics, University of Chile, Blanco Encalada 2002, Santiago, Chile
[2]GAIA-Antarctica, University of Magellanes, Avenida Bulnes 01855, Punta Arenas, Chile
[3]Center for Advanced Research in Arid Zones, Raúl Bitrán 1305, La Serena, Chile
[4]Niels Bohr Institute, University of Copenhagen, Blegdamsvej 17, Copenhagen, Denmark
[5]Department of Physics, University of Santiago de Chile, Avenida Ecuador 3493, Santiago, Chile
[6]Department of Physical Geography, Catholic University of Chile, Vicuña Mackenna 4860, Santiago, Chile

*Correspondence to:* Roland Eichinger (roland@dgf.uchile.cl)

**Abstract.** The Southern Ocean has been identified as a key player for the global atmospheric temperature and $pCO_2$ rise across the last glacial termination. One leading hypothesis for explaining the initial $pCO_2$ step of 38 ppm (Mystery Interval $17.5 - 14.5$ ka) is enhanced upwelling of Southern Ocean deep water that had stayed isolated from surface layers for millennia, thereby accumulating carbon from remineralisation of organic matter. However, the individual influences involved in this

interplay of processes are not fully understood. A credible explanation for this remarkable climate change must also be able to reproduce a simultaneous steep decrease of carbon isotope ratios ($\delta^{13}$C and $\Delta^{14}$C). To address this topic, we here apply the Danish Center for Earth System Science (DCESS) Earth System Model with an improved terrestrial biosphere module and tune it to a glacial steady-state within the constraints provided by various proxy data records. In addition to adjustments of physical and biogeochemical parameters to colder climate conditions, a sharp reduction of the oceanic mixing intensity below around

1800 m depth in the high latitude model ocean is imposed, generating a model analogy to isolated deep water while maintaining this water oxygenated in agreement with proxy data records. From this glacial state, transient sensitivity experiments across the last glacial termination are conducted in order to assess the influence of various mechanisms on the climate change of the Mystery Interval. We show that the upwelling of isolated deep water in the Southern Ocean complemented by several physical and biogeochemical processes can explain parts but not all of the atmospheric variations observed across the Mystery Interval.

# 1   Introduction

During the deglaciation after the Last Glacial Maximum (LGM, $\sim$21,000 years ago), global atmospheric temperatures rose by around 3.5 K (Shakun et al., 2012). At the same time, atmospheric $pCO_2$ increased, explaining parts of this temperature rise. Ice core records show a $pCO_2$ increase from 190 ppm during the LGM to Holocene conditions of 260 ppm in a series of steps (e.g. Monnin et al., 2001). On glacial-interglacial time scales, ocean processes largely determine $pCO_2$ variations and the Southern

Ocean (SO) has been found to be a key player for these variations (Fischer et al., 2010; Sigman et al., 2010). However, other physical and biogeochemical mechanisms in the ocean, atmosphere and on land also contribute to the overall climate change



and the individual contributions of the processes involved in this interplay remain unclear (Kohfeld and Ridgwell, 2009). A comprehensive explanation for the atmospheric $pCO_2$ rise across the last glacial termination is still lacking.

The most marked $pCO_2$ increase across the last glacial termination is a steep 38 ppm rise near its onset ("Mystery Interval" (MI), from 17.5 to 14.5 ka; Broecker and Barker, 2007). One leading hypothesis for explaining this initial step is enhanced upwelling and thus outgassing from the SO (Franois et al., 1997; Sigman and Boyle, 2000). This water had been isolated from surface layers and thereby accumulating carbon from remineralisation of organic matter during the glacial (Broecker and Barker, 2007). Such enhanced upwelling could result from a strengthening and/or poleward shift of the Southern Hemisphere west wind belt (Toggweiler and Russel, 2008; Anderson et al., 2009; d'Orgeville et al., 2010) and/or changes in ocean stratification through brine-induced effects on glacial-interglacial time scales (Bouttes et al., 2010, 2011; Mariotti et al., 2013). In addition to deep ocean upwelling, a number of other processes, such as variations in iron fertilisation of the SO, ocean volume changes, as well as carbon storage in the terrestrial biosphere (see e.g. Kohfeld and Ridgwell, 2009) and in permafrost (see e.g. Zech, 2012), have to be considered.

Recently, consistent records of carbon isotopes from ice cores have become available for use in constraining processes that control the global carbon cycle. Carbon isotope ratios are sensitive to variations in carbon exchange between the atmosphere, the terrestrial biosphere and the ocean reservoirs, but also to other effects on geological time scales such as changes in biogeochemical conditions. Therefore, analyses of the variations of isotope ratios can provide deeper insights into the mechanisms that are responsible for the $pCO_2$ and hence temperature increase. For example Reimer et al. (2013) and Schmitt et al. (2012) presented time series of reconstructions of the two carbon isotope ratios $\Delta^{14}C$ and $\delta^{13}C$ in the atmosphere during the past 25 ka. Both $\Delta^{14}C$ and $\delta^{13}C$ show a relatively sharp drop during the MI. While $\Delta^{14}C$ continues to decrease after that until present day, $\delta^{13}C$ rises again during the Holocene after first stagnating for around 4 ka. Schmitt et al. (2012) suggest that $\delta^{13}C$ is significantly influenced by ocean temperature changes only during the stagnating phase. The sharp drop can be related to the outgassing of isolated deep waters and the regrowth of the terrestrial biosphere can account for the increase during the Holocene. $\Delta^{14}C$ is largely affected by its cosmogenic production rate (see e.g. Lal and Peters, 1967), but estimates of $^{14}C$ production rates indicate that air-sea carbon exchange has to be considered to explain the $\Delta^{14}C$ drop during the MI (Laj et al., 2004; Muscheler et al., 2004, 2005; Hain et al., 2014). Burke and Robinson (2012) used deep sea coral data to reconstruct concentrations of this radioactive tracer (Half-life: $T_{1/2}(^{14}C) \approx 5730\,a$) in different oceanic water masses. Isotopically strongly depleted water masses can be found in the deep and intermediate SO (Sikes et al., 2000; Skinner et al., 2010; Thornalley et al., 2011). Due to the slow radioactive decay of $^{14}C$, isolation of these water masses from surface waters can account for this isotopic depletion during the last glacial. During the onset of the last glacial termination, $\Delta^{14}C$ in deep ocean waters rapidly increased, which indicates a sudden event of deep ocean mixing, transporting isotopically heavy waters into the deep ocean regions and hence also supporting the hypothesis of enhanced upwelling (Schmitt et al., 2012). Thereafter, deep ocean $\Delta^{14}C$ does not change significantly until the early Holocene. This indicates that the increase of atmospheric $CO_2$ in the latter part of the deglaciation is not primarily driven from the deep ocean.



Nevertheless, several studies call into question enhanced oceanic upwelling as the conclusive reason for atmospheric $CO_2$ and temperature variations during the MI. Due to limitations of the volume of an abundant isolated ocean reservoir, Broecker and Barker (2007) doubt that the magnitude of such an event could account for the observed atmospheric $CO_2$ changes. Mainly because of the lack of low $\Delta^{14}C$ in their measurements De Pol-Holz et al. (2010), Broecker and Clark (2010) and Cléroux

et al. (2011) question the existence of such strongly depleted ocean reservoirs. Moreover, modelling work by Hain et al. (2011) report shortcomings of the isolated reservoir hypothesis, amongst other reasons the study claims that the isolation causes widespread anoxia in the deep ocean (which was not reported by proxy data, see e.g., Jaccard et al., 2014) and that the carbon signal would rapidly be dissipated and diluted in the rest of the ocean. Other studies again, question the SO to be the origin of the event. Okazaki et al. (2010) and Rose et al. (2010) argue that North Pacific water masses could account for the observed $CO_2$

changes while Kwon et al. (2012) point out the possible influence of the North Atlantic.

A complete explanation for the $pCO_2$ increase at the onset of the last glacial termination must be able to reproduce a simultaneous decrease by 0.3‰ and 160‰ of atmospheric $\delta^{13}C$ (Schmitt et al., 2012) and $\Delta^{14}C$ (Reimer et al., 2013), respectively. Furthermore, it should also include how LGM deep water with high salinity, low $\delta^{13}C$ (Bouttes et al., 2011) and $\Delta^{14}C$ (Burke and Robinson, 2012) and low dissolved oxygen concentrations (but not widespread anoxia) was formed during the last glacial.

A complete understanding of the entire $pCO_2$ increase from 190 to 260 ppm across the last glacial termination requires the consideration of a globally comprehensive picture of the physical and biogeochemical processes in the atmosphere, the ocean and on land, as well as their interactions on various time scales.

New insights into these mechanisms can help to improve our understanding of global climate changes on centennial to millennial time scales and thereby allow more precise conclusions to be drawn about the crucial aspects that control the Earth's

climate system. For this reason, we here apply the Danish Center for Earth System Sciences (DCESS) Earth System Model (Shaffer et al., 2008), enhanced by a new terrestrial biosphere scheme with three vegetation zones and a permafrost component. We also develop a set of functions that describe the transitions of several parameters across the last 25 kaBP. This allows us to take a comprehensive approach, rather than concentrating on specific mechanisms (e.g. Bouttes et al., 2011; Tschumi et al., 2011; Chikamoto et al., 2012). Hitherto, the DCESS model has been used for future climate projections (see e.g. Shaffer et al.,

2009; Shaffer, 2010) and evaluated for pre-industrial (PI) climate conditions (see Shaffer et al., 2008). In the present study, it is calibrated for glacial conditions by adapting physical and biogeochemical parameters guided by proxy data records. This includes the imposition of a physically reasonable depth profile for the vertical exchange intensity in the high latitude model ocean to generate a model analogy to isolated deep and intermediate ocean waters. Transient sensitivity simulations across the last 25 kaBP are then performed. These demonstrate the impact and timing of various oceanic, atmospheric and terrestrial

processes on atmospheric temperatures, $pCO_2$ and the carbon isotopes $^{13}C$ and $^{14}C$ across the dynamic MI at the beginning of the last glacial termination.



## 2   Model description and new developments

The DCESS model features components for the atmosphere, ocean, ocean sediment, land biosphere and lithosphere and has been designed for global climate change simulations on time scales from years to millions of years (Shaffer et al., 2008). Its geometry consists of one hemisphere, divided into two 360° wide zones by 52° latitude. The model ocean is divided into a low-mid and a high latitude sector and features a continuous vertical resolution of 100 m, to a depth of 5500 m. The near surface atmospheric mean temperature is described by a simple, zonal mean, energy balance model in combination with sea ice and snow parameterisations. The atmosphere is assumed to be well mixed for gases and air-sea gas exchange fluxes and transports via weathering, volcanism, interactions with the land biosphere and anthropogenic activities are considered for carbon dioxide ($CO_2$) and methane ($CH_4$) in $^{12,13,14}C$ species, respectively, as well as for nitrous oxide ($N_2O$) and oxygen ($O_2$). Ocean dynamics are characterised by high latitude sinking and low-mid latitude upwelling as well as horizontal and vertical diffusion between the latitude zones and the ocean layers. For the ocean biogeochemical cycling, a number of tracers are considered (namely, phosphate ($PO_4$), dissolved oxygen ($O_2$), dissolved inorganic carbon ($DI^{12,13,14}C$), and alkalinity (ALK)), which are forced by new production, air-sea exchange, remineralisation of organic matter, dissolution of $CaCO_3$, river inputs and evaporation/precipitation (Shaffer, 1996; Shaffer et al., 2008). There is a sediment section for each of the ocean model layers addressing $CaCO_3$ dissolution and organic matter remineralisation.

A uniform land biosphere scheme accounts for the $^{12,13,14}C$ cycling with leaf, wood, litter and soil boxes (Shaffer et al., 2008). For the present study, we have extended this scheme to three different vegetation zones. We defined a latitudinal distinction of three different vegetation zones on a global scale: One for tropical forests (TF), a second for grasslands, savanna and deserts (GSD) and the third zone for extratropical forests (EF) containing temperate and boreal forests. The latitudinal limits of these vegetation zones are defined by the deviation of the global mean atmosphere temperature from its present day value. For this purpose, we derived two polynomial functions from a study by Gerber et al. (2004), where a complex vegetation model was applied to distinguish between a number of vegetation zones based on several variables. In addition, we chose the model "snowline" (latitude of 0°C annual mean temperature) as the poleward limit of the EF zone (but see below for the case of extensive ice sheets). Moreover, we included a representation of carbon, including $^{13}C$ and $^{14}C$, that is being trapped in the permafrost as well as below terrestrial ice sheets at glacial conditions and released during the deglaciation, which was not considered in the old version. Details about these new developments and an evaluation of the new module are given in the Supplement.

In the following, a set of physical and biogeochemical parameters is presented that serves to calibrate the DCESS model to glacial conditions guided by proxy-data records. This includes a reasonable vertical diffusion profile which constitutes a conceptual way to generate isolated deep water in the high latitude model ocean. Furthermore, we describe transition functions for a number of parameters used in the transient simulations across the last 25 kaBP.



## 2.1 Model Last Glacial Maximum

The calibration of the DCESS model to glacial climate conditions requires some developments, adaptations and adjustments. Guided by proxy-data records, we modified several biogeochemical and physical parameters to generate a model steady-state which fulfils all possible constraints for representing the LGM.

Firstly, in order to generate an analogy to isolated deep water in the SO as it has been described e.g. by Watson and Naveira Garabato (2006), the ocean vertical diffusion in the high latitude sector of the DCESS model ocean was modified. For PI climate conditions, the vertical exchange is set to $D_{PI} = 2.3 \cdot 10^{-3}\,\mathrm{m^2 s^{-1}}$, evenly throughout this model sector. To generate LGM conditions, we impose an ocean depth-dependent transition function on this parameter, a function that describes a sharp decrease in vertical diffusion at around 1800 m ocean depth until a background minimum is reached. Fig. 1 shows this

profile of the vertical diffusivity as function of ocean depth.

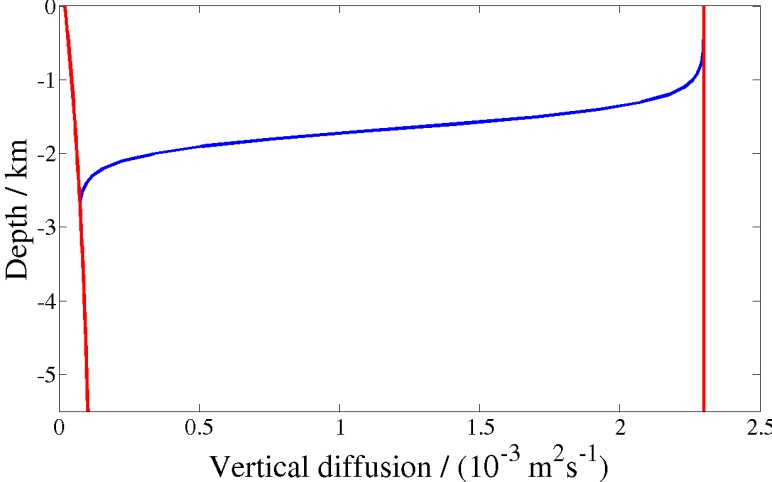

**Figure 1.** Depth profile of the vertical diffusivity in the high latitude model ocean for generating isolated intermediate and deep waters (blue line, bounded by red lines). The left red line denotes the background vertical mixing from the low latitude model ocean, which serves as a lower boundary and the right red line is the standard PI diffusion and describes the upper boundary for the LGM profile.

This imposes a sharp reduction of the high latitude ocean vertical diffusion and thus limits exchange of the upper ocean layers with intermediate and deep ocean waters. A detailed description of this profile is given in the Supplement. We varied the depth of the profile to obtain LGM climate conditions that constrain all required oceanic and atmospheric variables. Through the application of this diffusivity profile, the isolated ocean waters below the transition change towards high dissolved inorganic

carbon (DIC) and alkalinity values as well as towards low oxygen concentrations and $^{13,14}$C isotope ratios. In consequence of these oceanic changes only, the model PI climate atmosphere reacts with a pCO$_2$ drawdown of around 40 ppm and a 0.6 °C reduction of global mean atmosphere temperature. Furthermore, atmospheric $\delta^{13}$C increases by around 0.2‰ and $\Delta^{14}$C by around 50‰.





In addition to this variation in ocean dynamics, the generation of glacial climate conditions also requires the adaptation of biogeochemical parameters. For this, a number of parameters can be considered as possible candidates (see e.g. Kohfeld and Ridgwell, 2009). However, under consideration of the given possibilities the model provides and the knowledge about the respective parameter, the following adaptations have been conducted:

During the LGM, the dust concentration in the atmosphere was higher than during the Holocene (see e.g., Mahowald et al., 1999, 2006b; Maher et al., 2010), particularly in the high southern latitudes (see e.g. Lambert et al., 2013, 2015). Enhanced dust deposition over the ocean increases the iron supply and thereby the fertilisation in the ocean (Martin et al., 1990). Carbon export reconstructions presented by Lamy et al. (2014) and Martínez-García et al. (2014) confirm this enhancement of paleoproductivity in the SO. Hence, new production of organic matter through carbon remineralisation in the high latitude ocean

was higher during the LGM. In order to account for this effect, we modified the iron-limitation factor for new production in the high latitudes. Model tests yielded that a modification of the high latitude iron-limitation factor for new production from 0.36 (standard value for PI conditions, see Shaffer et al., 2008) to 0.5 can account for a productivity increase of around 40%, which seems to be a reasonable estimate for the entire area of the SO. This induces an atmospheric $pCO_2$ reduction of around 20 ppm, consistent with the DCESS model iron fertilisation results in Lambert et al. (2015).

During the LGM, the sea level was around 130 m lower (see e.g., Waelbroeck et al., 2002; Lambeck et al., 2014) than at present. This leads to exposed shelves around the continents and a reduced ocean volume. The difference of ocean volume between PI and LGM conditions is around 3.5% (see e.g. Adkins and Schrag, 2002). In the model, this is accounted for by keeping the phosphate inventory, which constitutes the nutrient limiting source in the DCESS ocean biochemistry, at a 3.5% higher level than for PI conditions. The reduced ocean volume also leads to enhanced ocean salinity during the LGM. We

therefore increased the mean model ocean salinity from the PI value 34.7 psu to 35.9 psu (see Adkins et al., 2002) to adjust it to glacial conditions.

To generate LGM conditions for $\Delta^{14}C$ in atmosphere and ocean, we applied the cosmogenic production $^{14}C$ rate of shortly before the LGM. This production rate is determined by the strength of the Earth's magnetic field, which shields the atmosphere from high energy cosmic particles, and the solar modulation of the incidence of high energy cosmic particles (Lal and Peters,

1967; Hain et al., 2014). For generating glacial conditions, the average from 25 to 26 kaBP of the $^{14}C$-cosmogenic production rate time series provided by Hain et al. (2014) was used. This corresponds to $PR_{14C} = 2.1 \cdot 10^4 \, \text{atoms/cm}^2\text{s}$.

Additionally, the equatorward expansion of terrestrial ice sheets is set to 47° latitude for generating LGM conditions. This is within the uncertainty range of LGM climate reconstructions (see e.g. Peltier, 2004) and has to be understood as a global two hemisphere average. The equatorward displacement of the model snowline and the expansion of terrestrial ice sheets comprise

three important climate feedbacks in this DCESS model version. Firstly, they determine the calculation of the global land albedo, which invokes a positive feedback to the cooling. Secondly, they limit the poleward expansion of the EF vegetation zone. This decreases the carbon uptake of the vegetation and thus leads to an atmospheric $pCO_2$ increase meaning a negative feedback to the cooling. Thirdly, another positive feedback to the cooling is generated through the storage of carbon in the increased land area below the ice sheets by the permafrost parameterisation in the new vegetation scheme.





The radiative forcing (RF) in the model follows the approximation by Budyko (1969)

$$RF = A - B \cdot T_{glob}, \tag{1}$$

where $T_{glob}$ denotes the global mean atmosphere temperature (in $°C$) and A and B parameters to adjust the climate sensitivity. In order to generate a climate sensitivity of $2.5°C$ (as proposed by Schmittner et al., 2011, for LGM simulations), we apply the longwave radiation to temperature sensitivity $B = 2.21\,\mathrm{Wm^{-2}K^{-1}}$ and the integrated mean incoming shortwave radiation $A_o = 207.16\,\mathrm{Wm^{-2}}$. $A = A_o - A_t$ then includes $A_t$, the sum of the radiative effects of carbon dioxide, methane and nitrous oxide. Data reconstructions presented by Schilt et al. (2010) show atmospheric partial pressures of around $pCH_4 = 380\,\mathrm{ppm}$ and $pN_2O = 200\,\mathrm{ppm}$ during the LGM. For the radiation calculations, we adopt these values. Moreover, we consider the radiative effect of additional atmospheric dust during the LGM (see above) by adding $A_{Dust}$ to $A_t$. According to Mahowald et al. (2006a), this accounts for an additional radiative effect of $-1\,Wm^{-2}$ for glacial conditions. An overview of all model parameters that were modified additionally to the ocean vertical diffusion is provided in Tab. 1.

| Adjusted parameters | PI | LGM |
|---|---|---|
| Fe-limitation factor | 0.36 | 0.5 |
| Phosphate inventory factor | 1 | 1.035 |
| Ocean mean salinity | 34.7 | 35.9 |
| Dust radiative forcing | $0\,\mathrm{Wm^{-2}}$ | $-1\,\mathrm{Wm^{-2}}$ |
| $^{14}$C production rate | $1.6\,\mathrm{atoms \cdot cm^{-1}s^{-1}}$ | $2.1\,\mathrm{atoms \cdot cm^{-1}s^{-1}}$ |
| $pN_2O$ for radiation | $270\,\mathrm{ppm}$ | $200\,\mathrm{ppm}$ |
| $pCH_4$ for radiation | $700\,\mathrm{ppm}$ | $380\,\mathrm{ppm}$ |
| Snow- / Iceline | $\sim 55°$ | $47°$ |

**Table 1.** Adjusted parameters for the generation of DCESS model LGM climate conditions.

When all these adaptations are applied, an 80 ka DCESS simulation leads to a steady climate state with conditions close to data-based LGM reconstructions. Atmospheric $pCO_2$ decreases to $189.2\,\mathrm{ppm}$ and the global mean atmosphere temperature to $11.52\,°C$. Moreover, atmospheric isotope ratios of $\delta^{13}C = -6.34‰$ and $\Delta^{14}C = 421.2‰$ are achieved. The ocean profiles for LGM conditions of various variables for the high and the low-mid latitude sector are presented in Fig. 2. For reference, also observed present low-mid latitude ocean profiles (Shaffer et al., 2008) are included in the figures.

A prominent feature in these vertical ocean profiles is generated through the transition of the vertical diffusivity to very low values below around 1800 m depth. Below the transition, the ocean shows high DIC concentrations and strongly depleted $^{13}$C isotope ratios (see Curry et al., 1998; Mackensen et al., 2001). It also shows low oxygen concentrations with minima of 0.05 mol m$^{-3}$ in the deep ocean in average. This indicates that there is no widespread anoxia in the ocean and is also in accordance with proxy data (Jaccard et al., 2014). Even though the isolated deep ocean waters are strongly $^{14}$C-depleted compared to the upper





**Figure 2.** Profiles of the DCESS model ocean at LGM conditions after $80\,ka$ of integration. Blue: high latitude model ocean profiles; red: low-mid latitude ocean profiles; black: observed present low-mid latitude ocean profiles (Shaffer et al., 2008).




ocean at LGM model conditions, deep ocean $\Delta^{14}$C values are still around $150\%o$ higher than at PI conditions. However, since atmospheric $\Delta^{14}$C has risen by more than $400\%o$ from PI to LGM conditions, the difference between atmospheric and oceanic $\Delta^{14}$C is much higher during the LGM. $max(\Delta\Delta^{14}$C$)$ (the maximum of $\Delta^{14}$C differences between atmosphere and ocean) is around $-480\%o$ during the LGM and also agrees well with observations (Burke and Robinson, 2012). Tab. 2 provides an overview of the result of key model variables of atmosphere and ocean for PI and LGM conditions as well as the proxy data LGM values for comparison.

| variable | Model PI | Model LGM | Proxy data LGM | Reference |
|---|---|---|---|---|
| $T_{atm}$ | $14.78°$C | $11.52°$C | $11.5$ - $11.8°$C | Shakun et al. (2012) |
| $pCO_2$ | $278\,\mathrm{ppm}$ | $189.2\,\mathrm{ppm}$ | $186 - 198\,\mathrm{ppm}$ | Lüthi et al. (2008) |
| $\delta^{13}C_{atm}$ | $-6.4\%o$ | $-6.36\%o$ | $-(6.38$ - $6.46)\%o$ | Schmitt et al. (2012) |
| $\Delta^{14}C_{atm}$ | $4\%o$ | $421.2\%o$ | $400$ - $570\%o$ | Reimer et al. (2013) |
| $min(\delta^{13}C_{oce})$ | $0.2\%o$ | $-0.5\%o$ | $-(1.1 - 0.3)\%o$ | Mackensen et al. (2001) |
| $max(\Delta\Delta^{14}C_{oce})$ | $5\%o$ | $-480\%o$ | $\sim -500\%o$ | Burke and Robinson (2012) |
| $min(O_{2,oce})$ | $0.12\,mol \cdot m^{-3}$ | $0.05\,mol \cdot m^{-3}$ | low but not anoxic | Jaccard et al. (2014) |

**Table 2.** Atmospheric and oceanic model variables for PI and LGM climate conditions and LGM proxy data for comparison. The range of the proxy data values express their variability between 18 and 25 kaBP.

## 2.2 Transient transition functions

For transient simulations over the last 25 kaBP, we initialised the DCESS model with the LGM conditions described above. For the variables and parameters that are not calculated interactively by the model, we developed specific functions that describe their transition from LGM to PI climate conditions as realistically as possible. For this, we either use time series of data-based reconstructions for their prescription, or time- or temperature-dependent parameterisations that represent their variations for the period of the last 25 kaBP. Here, we present these functions and explain how they have been derived.

Atmospheric partial pressures of the greenhouse gases methane and nitrous oxide can not be simulated adequately enough in these simulations, hence $pN_2O$ and $pCH_4$ are prescribed from the time series compiled by Schilt et al. (2010). Also, we parameterise the extension of the terrestrial ice sheets, in order to ensure an accurate albedo feedback as well as expansion of the EF vegetation zone and retreat of permafrost. From the above described globally averaged ice sheet expansion until $47°$ latitude during glacial conditions, we impose the temporal retreat of the ice line to the disappearance of the ice sheets at $70°$ latitude during the Holocene. For this, we use a data set presented in Shakun et al. (2012) showing the Northern Hemisphere (NH) ice sheet expansion from 100% at the LGM to 0% at present day. We then linearly prescribe the latitude of the ice line from $47°$ at 100% ice sheet expansion to $70°$ at 0% for the actual time step. The crucial value for the calculations of the albedo and the EF vegetation zone expansion, however, is determined by the minimum of this ice sheet line and the model-calculated





snowline. Fig. 3 shows the ice sheet line (converted as described from the data by Shakun et al., 2012) and a snowline as calculated by one of the transient model simulations from the following section (the All_TF simulation). The thick parts of the lines mark the current minimum of the two functions, which is the value that is applied in the calculations at the actual time step. This means that in this simulation between 15 and 0 kaBP the ice sheet line has no influence on the results and due to the

5 transition of the minimum between the functions (here at around 15 kaBP) some results may show a kink here.

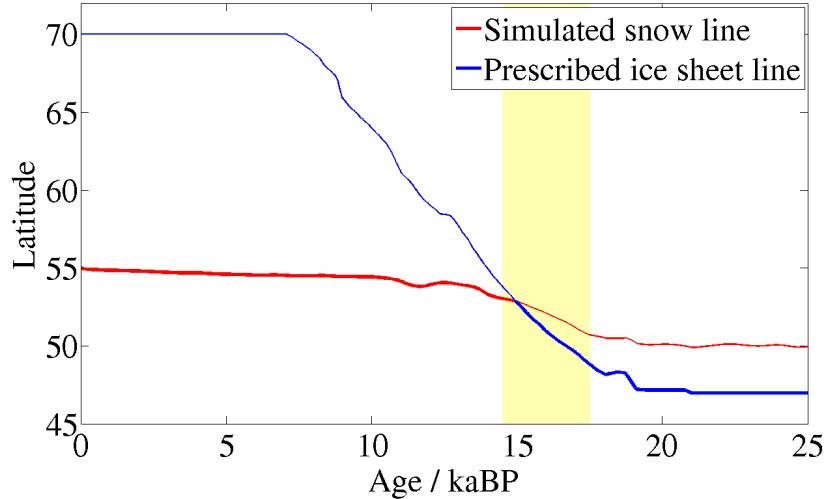

**Figure 3.** DCESS model-calculated "snowline" (red, from the All_TF simulation in Fig. 4) and prescribed line of NH ice sheet expansion (blue) converted as described in the text from data by Shakun et al. (2012). The current minimum of the two lines is used for the calculation of the actual albedo and vegetation extent. The yellow shading marks the period of the MI.

Due to ocean volume changes, the oceanic phosphate inventory was assumed to be at a constant level of 3.5% above the PI standard for generating LGM conditions. For the transition of this parameter across the last 25 kaBP, we linearly prescribe it to the latest sea level reconstruction time series from Lambeck et al. (2014). I.e. we associate a sea level of $-130$ m below present day values (as during the LGM) to a phosphate inventory scaled by $f_P = 1.035$. The temporal change of the sea level across

the last glacial termination then linearly determines the phosphate inventory so that $f_P = 1.0$ is reached, when the sea level is at present day values (0 m). Analogously, the ocean salinity is prescribed with a value of 35.9 psu for LGM sea level ($-130$ m) and 34.7 psu for present day (see Adkins et al., 2002).

As described above, the concentration of atmospheric dust influences the radiative budget, and by way of its deposition in the ocean, the iron limitation factor in the high latitude ocean sector. From proxy data records, we have developed a correlation

between the dust concentration and the global mean atmosphere temperature for the transition between LGM and Holocene. This yielded an exponential dependency of the dust concentration on the global mean atmosphere temperature that can be



described by

$$A_{Dust} = a \cdot e^{-0.4 \cdot T_{glob}} + b, \tag{2}$$

where a and b denote free parameters. Details about the derivation of this equation, the applied data and the transition functions that were derived for the radiative and for the iron fertilisation effect through this approach can be found in the Supplement.

As part of the computation of atmospheric $\Delta^{14}$C, the temporal variations of the cosmogenic $^{14}$C production rate have to be represented correctly in transient simulations. However, there are various approaches for how to establish a time series for this parameter; this is a subject of ongoing research. For most simulations in this study we use the most recent time series that has been developed by Hain et al. (2014). In Sect. 3.3, the $^{14}$C production rates from the studies by Laj et al. (2004) and Muscheler et al. (2004) are applied as well, in a sensitivity analysis. A description of the main characteristics of these data is given in the

Supplement.

    The main focus of this study is to test and quantify the hypothesis of a sudden event of deep ocean mixing, associated with the upwelling and outgassing of the isolated, carbon-rich SO waters during the MI. For this reason, we have developed a physically reasonable and simple method to resume the high, constant vertical diffusion of the PI high latitude model ocean between 17.5 and 14.5 kaBP. Starting at year 17.5 kaBP we modify the imposed function of the LGM vertical diffusion (see

Fig. 1) so that its transition towards very low values steadily decreases in ocean depth and hence mixing of the deeper ocean layers restarts. The velocity of this process is adjusted so that the full vertical exchange is recovered after around 3 ka in order to apply the entire effect of this process to the MI. Again, this should be understood as a model analogy to real SO processes that may have been at work during the MI climate transition. The technical description of the method is given in the Supplement.

## 3   Results of transient simulations

Here we present the results of the transient simulations across the last 25 kaBP. For these simulations, the model was initialised and forced with the conditions described in Sect. 2. Since this study focuses on the changes of atmospheric $CO_2$, temperatures, $\delta^{13}$C and $\Delta^{14}$C during the MI ($17.5 - 14.5$ kaBP) we here mainly present and discuss the global mean atmosphere values of these quantities from $20 - 10$ kaBP. First, we assess the impact of the various processes by successively applying the transition functions discussed above. Furthermore, the influences of permafrost and of the new biosphere scheme on the four quantities

are analysed through a sensitivity study. In the third part of the section, other $^{14}$C production rates are applied to investigate the influence of this rather poorly quantified effect and we calculate and compare the $^{14}$C production rate that is required to yield the $\Delta^{14}$C values from data-based observations. The section ends with a quantification and discussion of the results.


### 3.1 Analysis of individual processes

Four transient simulations have been conducted in order to assess the impacts of individual mechanisms on atmospheric $T_{glob}$, $pCO_2$, $\delta^{13}C$ and $\Delta^{14}C$ changes. First, only the prescription of the two minor greenhouse gases, methane and nitrous oxide, as well as the variations in ice sheet expansion are activated (simulation: Min_RF, **Min**or **r**adiative **f**orcing). In the second
5 simulation, the ocean vertical mixing is additionally restored as described above (Add_vdiff, **add**itional resumption of ocean **v**ertical **diff**usion). In the third simulation, variations in atmospheric dust concentrations are simulated on top of the other two effects through their impact on the radiative forcing and the iron limitation factor (Add_dust, **add**itional application of the **dust** transition functions). In the final simulation with all changes, the ocean phosphate inventory and the ocean salinity transition functions are applied as well (All_TF, use of **all t**ransition **f**unctions). The four panels in Fig. 4 show the four atmospheric
10 variables for these four simulations between 20 and 10 kaBP, including the MI.

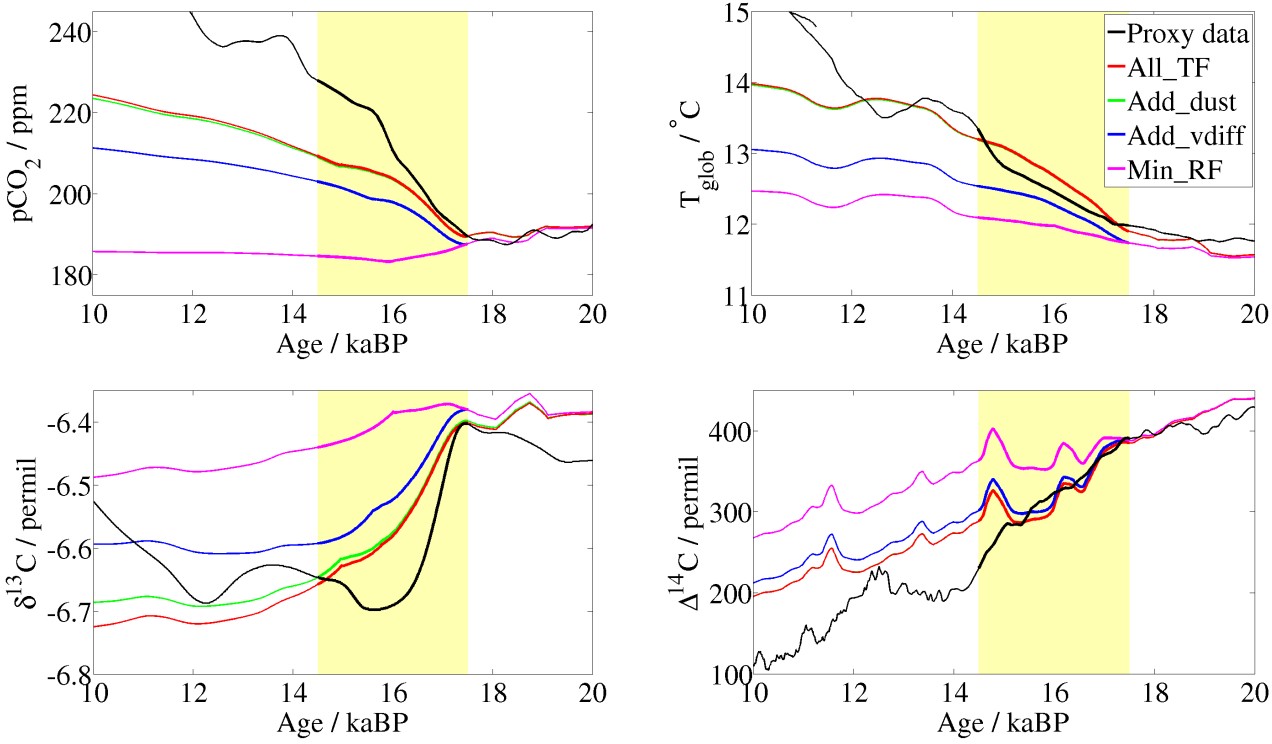

**Figure 4.** Atmospheric values for transient DCESS simulations from 20 to 10 kaBP and data-based reconstructions. Black: Data-based reconstructions; $pCO_2$ by Lüthi et al. (2008), temperatures by Shakun et al. (2012), $\delta^{13}C$ by Schmitt et al. (2012) and $\Delta^{14}C$ by Reimer et al. (2013); green: DCESS simulation only with prescribed ice sheet extent and minor greenhouse gas forcing (Min_RF); blue: as green, additionally with restored high latitude vertical exchange (Add_vdiff); magenta: as blue, additionally with prescribed temperature-dependent dust concentration (for radiative forcing and iron-limitation factor) (Add_dust); red: as magenta, additionally with sea level-dependent phosphate and salinity inventory variation (All_TF). The yellow shading indicates the MI from 17.5 to 14.5 kaBP.



As expected, changes in only methane, nitrous oxide and the ice sheet extent (Min_RF) have almost no influence on atmospheric $pCO_2$, and the temperature increase is also modest. The growth of the terrestrial vegetation through the prescribed ice sheet retreat even leads to a decrease in $pCO_2$ until about 16 kaBP. This is when the latitude of the ice sheets extends farther equatorward than the snowline (see Fig. 3, note that in the Min_RF simulation the two lines cross earlier because of

the slower increase of the snowline latitude). $\delta^{13}C$ and $\Delta^{14}C$ also show only minor changes, which are due to temperature and vegetation changes for $\delta^{13}C$ and mainly production rate-driven for $\Delta^{14}C$. The additional resumption of the ocean high latitude vertical diffusion (Add_vdiff) generates a $pCO_2$ change of around 18 ppm during the MI. Furthermore, $\delta^{13}C$ and $\Delta^{14}C$ drop within these 3 ka by an additional $0.15‰$ and $64‰$, respectively, due to the outgassing of the isotopically depleted deep waters. The dust-dependent iron limitation factor, as well as radiative forcing changes that are switched on in the third simulation

(Add_Dust) again enhance the changes of all four quantities during the MI. However, while the temperature and $\delta^{13}C$ change now already match the proxy data levels, $pCO_2$ and $\Delta^{14}C$ variations in the model simulation are still too small. The additions of the ocean phosphate and ocean salinity transition functions (All_TF) do not resolve this apparent shortcoming of the model simulation. These changes show their main, albeit limited, influence after the MI. In summary, our DCESS simulations successfully reproduce some aspects of the early last deglaciation, while others are still underestimated because important processes

are either missing or not adequately represented.

### 3.2 Impacts of permafrost and biosphere

The release of $CO_2$ and $CH_4$ into the atmosphere through the thawing of permafrost in a warming future climate has been assessed in a number of studies (see e.g. Schaefer et al., 2011; Schuur et al., 2008; Khvorostyanov, 2008). Carbon storage and release in/from permafrost can also help explain glacial-interglacial cycles (Zech, 2012). Hence, permafrost should be

considered when investigating the last deglaciation and thus has been parameterised in the newly implemented terrestrial biosphere scheme (see Supplement). In order to evaluate how much influence permafrost has on the atmospheric quantities considered here, we now deactivate the permafrost in the computation of the atmosphere-terrestrial biosphere fluxes (No_PF, **no** influence of **p**erma**f**rost). Thus, we set the additional permafrost-atmosphere carbon (including the rare isotopes) fluxes to zero throughout the simulation with all changes from the previous section. Fig. 5 shows the results of the All_TF simulation from

Sect. 3.1 and of the No_PF simulation without the effect of permafrost on the carbon exchange between land and atmosphere for all three carbon isotopes. Moreover, a simulation with the same conditions, but with the old land biosphere scheme, which does not feature the parameterisation for permafrost, Old_bio (use of the **old bio**sphere module) is included in this figure to evaluate the general influence of the vegetation zones and the permafrost on the simulation.

In the simulation without the influence of permafrost, a reduction of the change across the MI can be observed in all four

atmospheric variables compared to the reference simulation All_TF. The results of the two simulations start diverging at around 19 kaBP. This is when the change in ice sheet extent leads to first clear variations through its effect on the permafrost parameterisation in the model. Across the MI, the difference between the two lines increases strongly in all four panels and only slightly thereafter. Until the year 14.5 kaBP, the permafrost component accounts for an additional change of around 2.5 ppm in $pCO_2$ (6.5% of the 38 ppm change during the MI), about $0.1\,°C$ in global mean atmosphere temperature, only $-0.01‰$ in $\delta^{13}C$



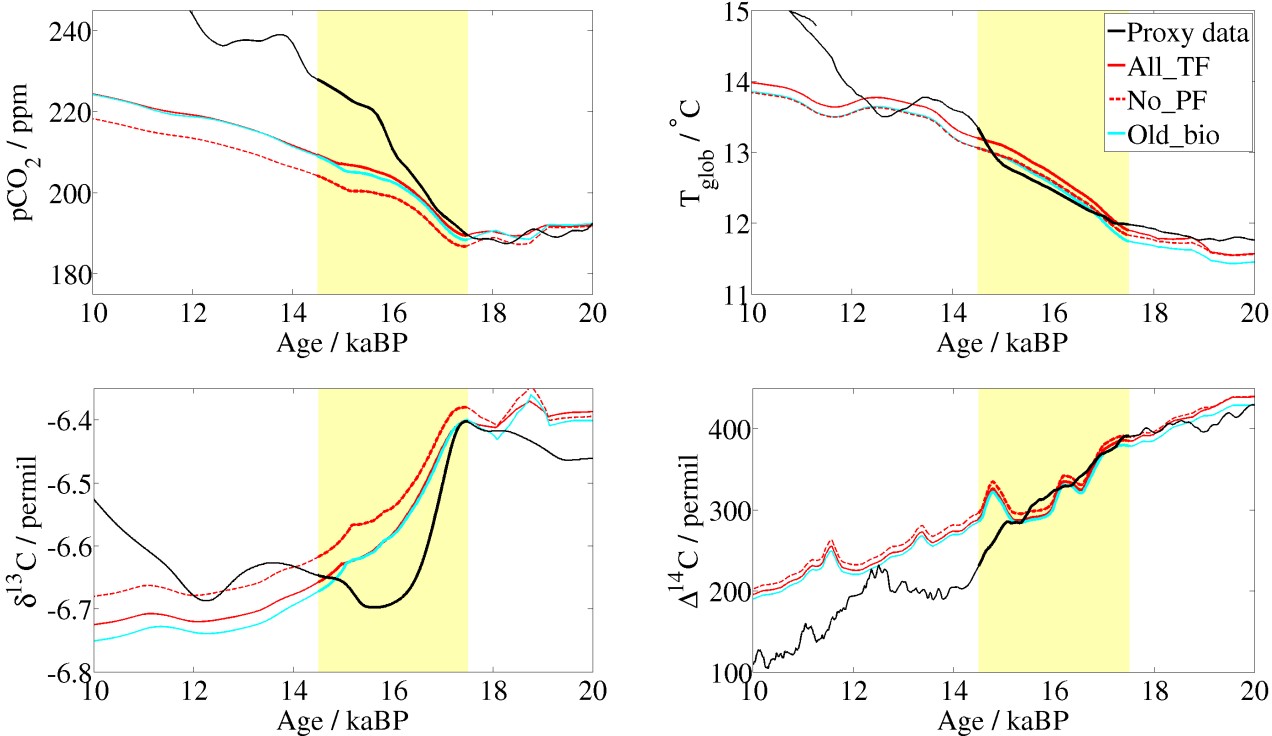

**Figure 5.** Atmospheric values for the DCESS simulation with all transition functions (All_TF, solid red line) and data-based reconstructions (black) as in Fig. 4. Dashed red line: DCESS simulation with all transitions functions, but deactivated permafrost component (No_PF). Light blue line: Same simulation with the old terrestrial biosphere scheme (Old_bio).

and $-3‰$ in $\Delta^{14}C$. This indicates, that for the overall climate change across the MI, permafrost plays only a secondary role, with only small contributions to temperature and isotopes (we prescribe carbon released from permafrost to be radiocarbon dead). The DCESS simulation without the newly implemented land biosphere scheme, i.e. without the distinction between the three vegetation zones and also without the consideration of permafrost, shows similar changes during the MI as in the All_TF
5    simulation for all variables. However, as shown in the Supplement, the new biosphere approach was needed to obtain the reconstructed land biomass decrease from PI to LGM and permafrost clearly plays an important role in global carbon cycling. Thus, both model enhancements are well motivated even if they do not yield "better" results in these particular simulations. Through the regrowth of the biosphere, particularly in the EF zone, atmospheric $CO_2$ is taken up by the vegetation. This compensates for a little more ($0.8\,\mathrm{ppm}$) than what is released through the parameterisation of permafrost carbon release. Since
10    $^{12}C$ is preferably taken up by the biosphere, also the $\delta^{13}C$ change during the MI is lower with the new biosphere scheme. $\Delta^{14}C$ is not sensitive to changes in vegetation and hence the influence of its release from permafrost dominates its development.





### 3.3 Evaluation of $^{14}$C production rates

As has been shown in Sect. 3.1, the change in $\Delta^{14}$C during the MI in the All_TF simulation is not as large as in the proxy data. Apart from atmospheric $CO_2$ itself and the release of deep ocean waters, $\Delta^{14}$C is strongly influenced by the cosmogenic production rate of $^{14}$C. This production rate is determined with rather large uncertainties and there are different ways to derive

it. In the Supplement, we present the three $^{14}$C production rate time series of the studies by Laj et al. (2004); Muscheler et al. (2004) and Hain et al. (2014) across the last 25 kaBP. Here, we present an evaluation of the three $^{14}$C production rate data applied to the simulation with all transition functions as described in Sect. 3.1. In the left panel of Fig. 6, we present the All_TF simulation for the three different production rates, as well as for a simulation with constant LGM-value production rate (Mus_PR, Muscheler et al. (2004) **p**roduction **r**ate; Laj_PR, Laj et al. (2004) **p**roduction **r**ate; LGM_PR, constant **LGM**-value

**p**roduction **r**ate). The proxy data record by Reimer et al. (2013) is also included in the figure.

    The simulation with constant $^{14}$C production rate at LGM level shows a $\Delta^{14}$C drop by 80‰ from the beginning to the end of the MI, almost entirely through the outgassing of isotopically depleted deep ocean waters. Neither of the $^{14}$C production rates can account for the remaining 80‰ reduction to explain the $\Delta^{14}$C decrease of 160‰ across the MI that can be seen in the data-based reconstruction by Reimer et al. (2013). With the data set by Hain et al. (2014), $\Delta^{14}$C drops by 96‰, using

the Laj et al. (2004) data, a 105‰ decrease can be explained and the Muscheler et al. (2004) time series only leads to $-58‰$ change. Furthermore, the proxy data does not show the production rate-caused variations within the MI and also, in the Mus_PR simulation, atmospheric $\Delta^{14}$C shows a large and sudden drop of around 150‰ shortly after the MI between 14.3 and 13.7 kaBP.

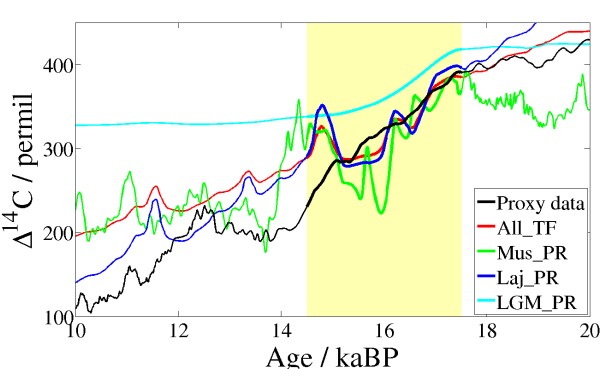
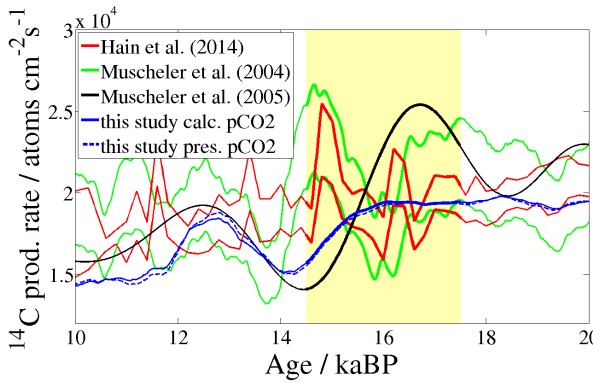

**Figure 6.** Left panel: $\Delta^{14}$C in transient simulations with all changes (see Sect. 3.1) applying different $^{14}$C production rates from Muscheler et al. (2004) (green, Mus_PR) Laj et al. (2004) (blue, Laj_PR) and Hain et al. (2014) (red, All_TF), and data based reconstructions from Reimer et al. (2013) (black). Right panel: Uncertainty ranges of the $^{14}$C production rate time series of the studies by Hain et al. (2014) (red) and by Muscheler et al. (2004) (green, processed with a 1000 a running mean), calculated $^{14}$C production rate from Muscheler et al. (2005) (black) and DCESS $^{14}$C production rate calculated to meet the $\Delta^{14}$C data by (Reimer et al., 2013) (blue solid: as All_TF simulation; blue dashed: with prescribed pCO$_2$ from Reimer et al. (2013), both processed with a 100 a running mean). The MI is indicated by yellow shading.





As for example shown in Muscheler et al. (2005), advanced assessment of this issue can be gained through the calculation of the $^{14}$C production rate that would be required to yield the $\Delta^{14}$C value from proxy data. In other words, we set the $^{14}$C production rate to 0 and prescribe atmospheric $\Delta^{14}$C to the data based reconstructions (from Reimer et al., 2013). The difference between the calculated model $\Delta^{14}$C and the data time series then yields the $^{14}$C production rate that would be needed to

generate the reconstructed $\Delta^{14}$C values in the model simulation. For comparison with this DCESS calculated $^{14}$C production rate, the right panel of Fig. 6 shows the uncertainty ranges of the data from the studies by Muscheler et al. (2004) and by Hain et al. (2014) and the $^{14}$C production rate calculated with a box diffusion model in the study by Muscheler et al. (2005). Due to the fact that the data of the study by Laj et al. (2004) is apparently outdated through its revision by Hain et al. (2014) (see Supplement), we did not consider that time series for this part of our study. We present this computation for one simulation

with all changes as presented in Sect. 3.1 and for one simulation with prescribed pCO$_2$ to provide an estimate for the influence of the model underestimated pCO$_2$ on the calculation of $\Delta^{14}$C.

Even though model-calculated and prescribed pCO$_2$ differ by up to 20 ppm during some periods of the simulation, the $^{14}$C production rate results of the two different model simulations are practically the same. This shows that changes of atmospheric $^{12}$C only weakly influence $\Delta^{14}$C compared to variations of $^{14}$C. At the beginning of the MI, the DCESS production rate still

agrees well with the data sets, but at around 15 kaBP, the DCESS time series drops while both data sets show an increase. Hence, the calculated production rate lies below both uncertainty ranges at that period, where also a large offset between observed and modelled $\Delta^{14}$C can be seen in Fig. 6. The box diffusion model calculated $^{14}$C production rate by Muscheler et al. (2005) also shows a strong decrease during this period where the observational data rises. After this peak, the observations decrease again and agree well with the DCESS calculations thereafter, particularly the time series by Muscheler et al. (2004). Across

the Holocene, the DCESS calculated $^{14}$C production rate is rather at the low end of the uncertainty ranges of the proxy data sets.

### 3.4 Discussion of results

The results of the transient DCESS model simulations across the last 25 kaBP demonstrate that some parts of the climate change at the onset of the last glacial termination can be explained by a combination of several physical and biogeochemical

processes and that upwelling of carbon-rich and isotopically depleted deep SO waters plays an important role in this interplay of processes. However, considerable aspects of the atmospheric changes can not be reproduced by the model and thus some mechanisms may not be represented adequately enough or are missing. In order to quantify the impact of individual processes on atmospheric changes during the MI in this modelling concept, Tab. 3 gives an overview of the differences of the four atmospheric variables between 17.5 and 14.5 kaBP in the model simulations presented herein and in proxy data records.

The model reproduces around 52% of the entire 38.4 ppm in atmospheric pCO$_2$ change (see Lüthi et al., 2008). To account for the rest, several processes can be thought of being insufficiently represented in the model and moreover, this could also be due to the timing of one or more of the transition functions, underrepresenting effects during the MI. E.g. Brovkin et al. (2007); Kohfeld and Ridgwell (2009) or Mariotti et al. (2013) discuss a number of processes that combined can account for the entire deglaciation, although with sometimes large uncertainties. We consider most of these processes to be accounted for in





| Change during MI | Δ pCO$_2$ / ppm | Δ Temp / °C | Δ $\delta^{13}C$ / ‰ | Δ $\Delta^{14}C$ / ‰ |
|---|---|---|---|---|
| Min_RF | −2.9 | 0.4 | −0.06 | −24.9 |
| Add_vdiff | 15.5 | 0.8 | −0.21 | −88.9 |
| Add_dust | 19.6 | 1.3 | −0.24 | −96.2 |
| All_TF | 19.9 | 1.3 | −0.25 | −96.4 |
| No_PF | 17.4 | 1.2 | −0.24 | −93.4 |
| Old_bio | 20.7 | 1.3 | −0.27 | −93.6 |
| Mus_PR | | | | −58.2 |
| Laj_PR | | | | −105.4 |
| LGM_PR | | | | −80.2 |
| Proxy data | 38.4 | 1.37 | −0.29[*] | −160.1 |

**Table 3.** Differences of atmospheric variables between the beginning (17.5 kaBP) and the end (14.5 kaBP) of the MI in various DCESS simulations and in data based reconstructions. [*]Since $\delta^{13}C$ increases again after a local minimum within the MI in the proxy data, we here provide the minimum value, rather than the value from the end of the MI (−0.24‰).

our simulations, but it is questionable if their influences during the MI in particular are correctly captured. Moreover, enhanced ocean remineralisation length scales during the glacial, due to less active bacteria at low temperatures, could trap more DIC in the deep ocean, which then could account for additional CO$_2$ outgassing but would also reduce deep ocean dissolved oxygen concentrations. Also, the glacial, due to the volume of deep ocean waters that upwelled during the MI, is very uncertain and water masses in other oceans may also have contributed to the overall change (Rose et al., 2010; Okazaki et al., 2010; Kwon et al., 2012). Furthermore, although considerably improved, the representation of the biosphere, its regrowth and uptake of atmospheric carbon as well as the parameterisation of permafrost bears a number of uncertainties. While the two-way impact of dust plus the thawing of permafrost and the other biogeochemical processes sum up to around 7 ppm pCO$_2$ change during these 3 ka, the upwelling of deep SO waters accounts for around 16 ppm pCO$_2$ change in the DCESS model simulation (together with ocean warming). In total, this still explains more than 40% of the complete MI pCO$_2$ change in observations and thus, the upwelling of deep SO waters has the potential to be a major driver for the early last deglacial change in atmospheric pCO$_2$.

The global mean atmosphere temperature change almost matches with the change in the data compiled by Shakun et al. (2012). This is despite the fact that the atmospheric pCO$_2$ change during the MI is not entirely simulated by the model and therefore not entirely realistic. The approximated climate sensitivity in the model could account for this, although the temperature-dependent transition function for dust also bears large uncertainties. The dust component accounts for about 0.5 °C global temperature change during the MI, and most of that is related to the direct dust radiative forcing. The temperature and pCO$_2$ changes after the MI across the Bølling Allerød, the Younger Dryas and the Holocene are not expected to be simulated





in detail by the DCESS model. Due to the model's simplified geometry with only one hemisphere, interactions between the hemispheres and thus the bipolar seesaw can not be represented.

In comparison with the data based reconstructions by Burke and Robinson (2012), the change in $\delta^{13}$C during the MI is only slightly underestimated by the model (by 0.04‰). Again, the main contribution to this strong decrease is caused by deep ocean

upwelling, which accounts for almost 60% of the overall change. Also Brovkin et al. (2002) point out the importance of oceanic processes for atmospheric $\delta^{13}$C at glacial-interglacial time scales. In general, the $\delta^{13}$C drop at the onset of the deglaciation can be seen as being well represented by this DCESS modelling approach, although, other features of the simulation suggest that some influence on atmospheric $\delta^{13}$C is lacking or not correctly described in the model code. Firstly, before the MI, $\delta^{13}$C is slightly increasing, while in the simulation it decreases, mainly due to the regression of the ice sheets and thawing of

permafrost. Secondly, after around 12 kaBP, $\delta^{13}$C increases again in the observations, while it remains relatively constant in the simulations. Burke and Robinson (2012) mainly attribute this rise to the continuing regrowth of the biosphere, which does not have such a strong effect on atmospheric $\delta^{13}$C in the model. The $\delta^{13}$C dip during the Bølling Allerød, on the other hand, is not expected to be prominent in the simulation, since this behaviour is thought to be mainly connected with ocean circulation changes that are not resolved in the model.

The much discussed sharp $\Delta^{14}$C drop of 160‰ (see Reimer et al., 2013) (note that in previous studies by Broecker and Barker (2007) or Reimer et al. (2009) this was referred to as 190‰) at the early last deglaciation can not entirely be explained by this modelling study. By applying a constant LGM $^{14}$C production rate, all the above described processes can account for an 80‰ change and about 53‰ of those can be attributed to the effect of the resumption of the high latitude vertical diffusion. Neither of the three different time series of the $^{14}$C production rate can account for the rest of the $\Delta^{14}$C change. At most,

the data by Laj et al. (2004) leads to an additional 25‰ decrease. However, the determination of the $^{14}$C production rate is obviously subject to large uncertainties. For example, the drop in the Muscheler et al. (2004) time series at around 14 kaBP leads to a sudden 150‰ decrease in $\Delta^{14}$C in our model simulation but can not be seen in $\Delta^{14}$C proxy data. This allows room for speculations about timing inaccuracies in the data set. In this context, it should be mentioned, that recent revisions to ice core time scales have not yet been applied for revising the reconstructed snow accumulation rates and $^{10}$Be fluxes and its

influence on the $^{10}$Be-based $^{14}$C production rate (R. Muscheler, personal communication, 2015). The impact of permafrost on $\Delta^{14}$C is very small, even though we assume carbon released from permafrost to be radiocarbon dead. The expected radiocarbon decrease generated through permafrost thawing can apparently be compensated by ocean-atmosphere exchange and subsequent mixing to the deeper ocean.

The model-calculated $^{14}$C production rate lies within the uncertainty limits during most of the simulation but shows large

deviations at the end of the MI. This discrepancy can also be observed in the box diffusion model calculated time series by Muscheler et al. (2005) (which assumes a constant carbon cycle). Hence, the question has to remain open if it is the lack of an important process in the model, which could possibly also account for the remaining increase of atmospheric $pCO_2$, or errors in the production rate data sets that lead to these deviations.

In summary, it can be stated that our configuration of the DCESS model with the newly implemented land biosphere scheme

is capable of simulating some important aspects associated with changes in climate and carbon cycling during the period 17.5





to 14.5 kaBP. However, some variations of atmospheric pCO$_2$, temperature, $\Delta^{14}$C and $\delta^{13}$C as seen in proxy data records can not be reproduced by our model study. These remaining changes could possibly be captured by applying a dynamically more complex model, or by revising and/or adding one or more model parameterisations.

## 4   Conclusions

This study constitutes to our knowledge the first transient earth system modelling approach to investigate the onset of the last glacial termination using model-data comparisons of atmospheric temperatures, pCO$_2$, $\delta^{13}$C and $\Delta^{14}$C concurrently. Along with a number of established adaptations of parameters to glacial climate conditions, the DCESS model successfully simulates Last Glacial Maximum conditions in the ocean and atmosphere when a sharp reduction of the high latitude vertical diffusion is imposed below 1800 m ocean depth thereby generating isolated deep water. Transient model simulations show that the re-

sumption of the ocean vertical exchange and a thereby induced outgassing of carbon-rich and isotopically depleted deep ocean waters, can account for large parts of the exceptional change in atmospheric pCO$_2$, $\delta^{13}C$, $\Delta^{14}C$ and T$_{glob}$ at the onset of the last glacial termination (Mystery Interval, 17.5-14.5 kaBP) shown in various proxy data records. Although the temperature and the $\delta^{13}$C variations are simulated in accordance with proxy data by the model, the proxy-derived CO$_2$ and $\Delta^{14}$C changes across the Mystery Interval can only partly be reproduced. The thawing of permafrost due to atmospheric warming and retreat

of ice sheets, as well as the regrowth of the terrestrial biosphere, is shown to play a moderate role in explaining the climate change of this period of the last deglaciation, particularly for pCO$_2$ and $\delta^{13}C$. The assessment of the impact of various $^{14}$C production rates on the model results can not conclusively explain the well-known "mysterious" drop of $\Delta^{14}$C at the onset of the last deglaciation, but provides new insights to the impacts of the individual mechanisms controlling this behaviour. This study shows that the hypothesis that a sudden event of upwelling of isolated deep waters of only the Southern Ocean does

not suffice to account for the entire climate change of the Mystery Interval, however, it may be a main contributor to that change.

Supplementary material related to this article is available online at doi:10.5194/cp-0-1-2016-supplement.

*Acknowledgements.* We thank Raimund Muscheler for providing $^{14}$C production rate data and information about it as well as Ricardo De Pol-Holz for discussions. This work was financed by Chilean Nucleus NC120066. GS and NA acknowledge support by FONDECYT grants

# 1120040 and # 1150913, MR by Fondecyt grant # 1131055, and FL by Fondecyt grant # 1151427.





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
