# Peer review of "A model-data assessment of the role of Southern Ocean processes in the last glacial termination"

_Climate of the Past, 2015_

## Referee Comment (RC1) · Anonymous Referee #1 · 24 Feb 2016

Eichinger et al. present results of simulations performed with DCESS model of intermediate complexity. The main goal of the study is to understand the changes in atmospheric D14C between 17.5 and 14.5 ka B.P. The impact of changes in permafrost, deep ocean ventilation, dust and a higher PO4 content are briefly looked into over the whole deglaciation. My main criticism is that in some ways the authors try to do too much without really looking at each of the processes. So, at the end little new information is coming out of the manuscript or some important information is missing to really understand the implications. Please find some specific comments below.

The model used here is a simple Earth System Model of intermediate complexity, which comprises one high latitude zone and one low latitude zone, without "proper" water

[Figure]

masses. This is thus a very idealized (simple) set up, which has significant implications when discussing the deglaciation, changes in permafrost, upwelling...

1. Southern Ocean upwelling:

The authors always refer to Southern Ocean upwelling, whereas there is no "proper" (NADW, AABW...) water masses. In addition, in the pre-industrial set up water down-wells in the high latitude box and upwells in the low latitude box. There is therefore no Southern Ocean upwelling (as far as I can tell, because there is little information on the topic). In the LGM state, vertical diffusion is reduced in the high latitude box below 1km. Therefore in the LGM state sinking at high latitude is restricted to ∼2000m, correct?

What are the water transports at the PI and LGM?

The change in ventilation is simulated by a change in vertical diffusion, whereby diffusion linearly increases between 17.5 and 14.5 ka B.P. So is the upwelling during MI occurring in the low latitude box in the model? Is the whole deep ocean ventilated between 17 and 15 ka B.P.? What are the associated changes in water transport during MI?

Shouldn't the authors show the vertical profiles of oceanic D14C, d13C and DIC, ALK and O2 at ∼ 15 ka B.P? The alkalinity profiles are not shown whereas it plays a strong control on atm. pCO2. All this information is crucial to understand the physical and biogeochemical response of the model to the forcing. The implications associated with the experimental design and model geometry ishould be clearly discussed and should be compared thoroughly with previous studies (see also references issues).

2. Heinrich 1-B/A:

Throughout the manuscript the authors refer to the "Mystery Interval", while totaling ignoring what we could call the "climatic intervals" of the last deglaciation,(Heinrich 1, the Bolling-Allerod ) and the possible changes in oceanic circulation which occurred during that time period. While these changes in oceanic circulation cannot be simulated

with the model used here, the implications should be discussed. H1 is not a simple case of "enhanced Southern Ocean upwelling" as NADW was probably very weak.

3. Given these simplifications, the calculated C14 production rate is also associated with high uncertainties, even though it is quite informative to use several C14 production rate forcings. This is one of the most interesting part of the paper. As a summary, the experiments presented here serve as an estimate of the impact of high latitude diffusion change on atmospheric D14C. In general simple models give an upper estimate of possible changes, and if really the whole ocean below 2000m gets ventilated during MI, you would expect the value presented here to be an upper estimate. . .

4.Permafrost: The authors briefly study the impact of changes in permafrost across the deglaciation on pCO2, D14C and d13C, but some important information is missing. How much permafrost is stored at the LGM compared to PI? How does that compare with other studies? What is the time evolution of the permafrost changes? Changes in permafrost are associated with high uncertainties and given the very simple approach used here, a more accurate change in terrestrial carbon content associated with permafrost cannot be obtained here. The experiment can however give a bit of information on the associated change in pCO2 and D14C for a given terrestrial carbon release, but it is imperative to know the change in terrestrial carbon (in GtC) separated into vegetation and permafrost. There is no discussion on the reasons behind the possible changes in permafrost, their timing . . ... As such I don't really see the added value of the permafrost section. Also please see Kohler et al. 2014 (Nat. Comm.).

5. References: The introduction is imprecise and lacks a lot of important references for the work described. A lot of work on glacial/interglacial changes in pCO2 is not mentioned (see work by F. Joos team for example). A lot of references on C14Âăwork is missing (Kohler et al. 2014, Huiskamp and Meissner 2012).

Some references are also used in the wrong context:

For example p 18, the reference to Burke and Robinson (2012) is completely misused

as they present D14C data from the Southern Ocean and not d13C data. As such the whole paragraph p18, L3-14 is wrong and please note that there is no d13CO2 dip during the B/A.

P3, L9-10 in the intro is wrong. P6, L9 is wrong

―――――――――――――――

---

## Referee Comment (RC2) · Anonymous Referee #2 · 3 Mar 2016

This study uses a simplified numerical climate model to assess the role of different processes on the last glacial termination. This is an important topic for which many questions remain. Although it is a good idea to combine data and model to better understand the changes taking place at the termination, I am not sure the tool used here is suited for the task. The model, including new changes and shortcomings due to its simplicity, should be better explained. Other studies have focused on the last deglaciation with more complex models, this paper should better explain what is new here compared to previous work. It might need to be re-written in a clearer way.

General comments

- Model used: I have serious concerns regarding the suitability of the numerical model

used: it's a simplified model with only one hemisphere, a simple atmosphere (EBM) and no real ocean dynamics. The terrestrial biosphere model only depends on temperature (not precipitation). Is it enough to draw conclusions on changes impacting the carbon cycle on Earth based on changes in terrestrial biosphere and ocean dynamics?

- Methods: Several changes have been made on the model. They seem important for the study and should not be in the supplementary material, but in the main text as they are relevant to the results.

- Novelty: what is new in this study? The permafrost part has not been studied before, but this is not the main topic of the study (the Southern Ocean) and the permafrost module seems very simple and is not validated. The other mechanisms have already been studied in the past with better suited models, especially changes in ocean dynamics with models that better simulate the ocean dynamics (Tschumi et al., 2011; Bouttes et al., 2012; Brovkin etal., 2012; Menviel et al., 2012; Mariotti et al., 2016). The most interesting and new part is probably the section on carbon 14 and the role of the production rate, but then the paper should be re-organised around this, and the new work by Mariotti et al. (2016) discussed.

- The main process that is studied is the change of mixing in the ocean. But the model-data comparison only focusses on atmospheric carbon isotopes. Since the main change comes from the ocean, it would be better to also compare model-data for carbon isotopes in the ocean.

Specific comments

In the abstract (and in other sections) the authors are very vague on the different processes and mechanisms studied such as p. 1 lines 4-5: "this interplay of processes" p.1 line 12: "various mechanisms" Could you be more precise?

Can you also be more precise concerning the variables you're looking at? P.1 line 14: "the atmospheric variations": what are you referring to?

p.1 line 21: "other [...] mechanisms": can you be more precise?

p.1 line 21: "also contribute to the overall climate change": are you talking about the climate change or the changes in the carbon cycle? The same issue arises several times in the text, there seems to be a confusion between carbon cycle and climate.

p.3 line 13: The reference for LGM high salinity should be Adkins et al., 2002, the one for d13C: Curry and Oppo (2005), not Bouttes et al. (2011).

p.3 line 22: We HAVE also DEVELOPED a set of functions...

p3 line 23: I'm not sure I agree with the fact that previous studies have looked at only one mechanism while this one would have a more comprehensive approach. I think the novelty of the study here compared to previous ones should be better explained, and specifically what is different here. Other models were already taking into account different mechanisms (ocean dynamics, biogeochemistry in the ocean, terrestrial biosphere...) with better suited models. The only thing that seems new to me is the inclusion of permafrost, but very few is said about it. Other similar work with models that should be discussed: Brovkin et al., 2012; Menviel et al., 2012; Mariotti et al., 2016. Rather than Bouttes et al. (2011), the comparison should be with the study on transient simulations (Bouttes et al., 2012).

p. 4 line 4: Given that this study aims at studying the role of Southern Ocean processes, isn't it an issue that the model has only one hemisphere?

p. 4 line 6: Is the use of an EBM sufficient to correctly represent the terrestrial biosphere changes during the termination?

p.4 line 8: what are the "anthropogenic activities" for the deglacial simulations? If it's not relevant for this study it should not be mentioned. Also how are the volcanism and weathering taken into account for this period?

p.4 lines 16-27: Does the definition of the three additional zones have any impact on the carbon cycle (amount of carbon stored, isotopic fractionation) and /or on climate

(albedo...) or is it just an output to compare with data? The new changes are presumably important for the study and might be what is new compared to others (especially the permafrost): the developments should be included in the text and not be put in the supplement.

p.5 line 3: specify which proxy-records

p.5 lines 5-10: could you explain the physical rationale to have such a diffusion profile with a sharp reduction with depth (apart from changing the mixing), i.e. why would the mixing be different at the LGM? Given that there is only one hemisphere, is it a problem that you do it for the entire high latitude ocean and not only the Southern Ocean?

p.5 line 12: Given that the change of the diffusion profile is the main process studied, it should be explained here and not in the supplement. In the supplement you do not explain what variables were used to find the best guess profile. This should be shown (in the main manuscript) with the comparison with data.

p. 6 lines 33-34: Explain the storage of carbon below the ice sheet and in permafrost and what data are used to constrain it.

p.7 lines 12-14: can you give the values from the data to compare with the model results?

Figure 2. Please add a,b,c... for each panel. Could you also add the legend (blue, red, black). The units should be given in brackets. As the goal is to compare with data, can you add the data for the variables for which they exist, such as the carbon isotopes?

Table 2 and in other places in the text: for oceanic d13C data use Curry and Oppo (2005), Hesse et al. (2011), Peterson et al. (2014).

Figure 3. Put units in brackets.

p.11 line 18: given that the change of diffusion is the main mechanism studied and most of the results come from its change, it should not be detailed in the supplement

but in the main text.

Figure 4. add a,b,c for each panel and units in brackets.

p.13 line 2: can you give the changes of carbon stored in the terrestrial biosphere? Is this in line with data and previous simulations? (e.g. Ciais et al., 2012)

Figure 5 and 6 . add a,b,c for each panel and units in brackets.

p.19 lines 5-6: this is not true anymore, see Mariotti et al., 2016.

References

Bouttes, N., Paillard, D., Roche, D. M., Waelbroeck, C., Kageyama, M., Lourantou, A., Michel, E., and Bopp, L.: Impact of oceanic processes on the carbon cycle during the last termination, Clim. Past, 8, 149-170, doi:10.5194/cp-8-149-2012, 2012.

Brovkin, V., Ganopolski, A., Archer, D., and Munhoven, G. (2012), Glacial CO2 cycle as a succession of key physical and biogeochemical processes, Clim. Past, 8, 251-264, doi:10.5194/cp-8-251-2012.

Ciais, P., A. Tagliabue, M. Cuntz, L. Bopp, M. Scholze, G. Hoffmann, A. Lourantou, S. P. Harrison, I. C. Prentice, D. I. Kelley, C. Koven and S. L. Piao (2012), Large inert carbon pool in the terrestrial biosphere during the Last Glacial Maximum, Nature Geoscience, 5, 74-79, doi:10.1038/ngeo1324

Curry, W. B. and D. W. Oppo (2005), Glacial water mass geometry and the distribution of $\delta$13C of $\Sigma$CO2 in the western Atlantic Ocean, Paleoceanography, 20, PA1017, doi:10.1029/2004PA001021.

Hesse, T., M. Butzin, T. Bickert, and G. Lohmann (2011), A model-data comparison of \u03b413C in the glacial Atlantic Ocean, Paleoceanography, 26, PA3220, doi:10.1029/2010PA002085.

Mariotti, V., D. Paillard, L. Bopp, D.M. Roche, and N. Bouttes (2016), A coupled model

for carbon and radiocarbon evolution during the last deglaciation, Geophys. Res. Lett., 43, doi:10.1002/2015GL067489.

Menviel, L., F. Joos, S.P. Ritz, Simulating atmospheric CO2, 13C and the marine carbon cycle during the Last Glacial–Interglacial cycle: possible role for a deepening of the mean remineralization depth and an increase in the oceanic nutrient inventory (2012), Quaternary Science Reviews, 56, 46–68.

Peterson, C. D., L. E. Lisiecki, and J. V. Stern (2014), Deglacial whole-ocean $\delta$13C change estimated from 480 benthic foraminiferal records, Paleoceanography, 29, 549–563, doi:10.1002/2013PA002552.

---

## Author Comment (AC1) · 15 Mar 2016

**Reply to:**
*on "A model-data assessment of the role of Southern Ocean processes in the last glacial termination" by R. Eichinger et al. (cp-2015-190)*
**from Anonymous Referee #1**

*Dear Anonymous Referee #1,*

*thank you very much for your valuable comments and suggestions. Please find our answers (in blue) to your comments (in black) below:*

Eichinger et al. present results of simulations performed with DCESS model of inter-mediate complexity. The main goal of the study is to understand the changes in at-mospheric D14C between 17.5 and 14.5 ka B.P. The impact of changes in permafrost, deep ocean ventilation, dust and a higher PO4 content are briefly looked into over the whole deglaciation. My main criticism is that in some ways the authors try to do too much without really looking at each of the processes. So, at the end little new informa- tion is coming out of the manuscript or some important information is missing to really understand the implications. Please find some specific comments below. The model used here is a simple Earth System Model of intermediate complexity, which comprises one high latitude zone and one low latitude zone, without "proper" water masses. This is thus a very idealized (simple) set up, which has significant implications when discussing the deglaciation, changes in permafrost, upwelling...

1. Southern Ocean upwelling:

   • The authors always refer to Southern Ocean upwelling, whereas there is no "proper" (NADW, AABW. . .) water masses. In addition, in the pre-industrial set up water downwells in the high latitude box and upwells in the low latitude box. There is therefore no Southern Ocean upwelling (as far as I can tell, because there is little information on the topic). In the LGM state, vertical diffusion is reduced in the high latitude box below 1km. Therefore in the LGM state sinking at high latitude is restricted to ∼2000m, correct? What are the water transports at the PI and LGM?

   The described ocean overturning is the same in LGM as in PI state, this is independent from the changes in vertical diffusion that we perform and not
relevant for our study. It is the change in vertical diffusion that generates our LGM state by isolating the water masses in the high latitude ocean. However, this vertical exchange intensity change should not be understood as a change in real vertical diffusion, but rather as a conceptual way of separating the water masses or a model analogy to isolated deep water generation. We will add some additional explanations to clarify this apparant misunderstanding of the concept.

- The change in ventilation is simulated by a change in vertical diffusion, whereby diffusion linearly increases between 17.5 and 14.5 ka B.P. So is the upwelling during MI occurring in the low latitude box in the model? Is the whole deep ocean ventilated between 17 and 15 ka B.P.? What are the associated changes in water transport during MI?

  No, the "upwelling" takes place in the high latitude box, the overturning is not responsible for that, it is the recovery of the PI vertical diffusion profile. The deep water in the high latitude sector that had been isolated during glacial state can now (through restoration of vertical exchange intensity) mix with the upper ocean waters again, promoting outgassing. This is intended as an analogy to the (much more complex) changes in real ocean dynamics during the MI. We will revise the manuscript to make this point more clear.

- Shouldn't the authors show the vertical profiles of oceanic D14C, d13C and DIC, ALK and O2 at ~15 ka B.P? The alkalinity profiles are not shown whereas it plays a strong control on atm. pCO2. All this information is crucial to understand the physical and biogeochemical response of the model to the forcing. The implications associated with the experimental design and model geometry should be clearly discussed and should be compared thoroughly with previous studies (see also references issues).

  After resumption of the ocean vertical diffusion, the ocean profiles go back toward the initial PI state shown in Shaffer et al. (2008). That state had been calibrated to reproduce observed low-mid latitude profiles, profiles that we

do show in our Fig. 2. Thanks for pointing this out. We will add the alkalinity profiles and will explain in more detail the changes taking place below and above the reduced vertical diffusion.

2. Heinrich 1-B/A:

- Throughout the manuscript the authors refer to the "Mystery Interval", while totaling ignoring what we could call the "climatic intervals" of the last deglaciation,(Heinrich 1, the Bolling-Allerod ) and the possible changes in oceanic circulation which occurred during that time period. While these changes in oceanic circulation cannot be simulated with the model used here, the implications should be discussed. H1 is not a simple case of "enhanced Southern Ocean upwelling" as NADW was probably very weak.

    Thank you for mentioning this, we will add some more discussion on this topic. Please note, however, that the goal of the present study is to investigate how much of the climate change during MI can be accounted for by Southern Ocean upwelling alone (+ other processes that can be represented by the model). In conclusion we also state that other (oceanic) mechanisms may be responsible for the rest of the climate change that could not be captured in our simulations, so we do not think there is a contradiction here.

3. Given these simplifications, the calculated C14 production rate is also associated with high uncertainties, even though it is quite informative to use several C14 production rate forcings. This is one of the most interesting part of the paper. As a summary, the experiments presented here serve as an estimate of the impact of high latitude diffusion change on atmospheric D14C. In general simple models give an upper estimate of possible changes, and if really the whole ocean below 2000m gets ventilated during MI, you would expect the value presented here to be an upper estimate. . .

As described above, please do not mistake the change in "vertical diffusion" performed here with literal oceanic vertical diffusion change, it is (just) a model analogy. Yes, this is representing an upper estimate of deep SOUTHERN Ocean ventilation during the MI, possible other water masses being ventilated are not captured of course. Thanks for pointing this out, we will add this point to the discussion.

4. Permafrost: The authors briefly study the impact of changes in permafrost across the deglaciation on pCO2, D14C and d13C, but some important information is missing. How much permafrost is stored at the LGM compared to PI? How does that compare with other studies? What is the time evolution of the permafrost changes? Changes in permafrost are associated with high uncertainties and given the very simple approach used here, a more accurate change in terrestrial carbon content associated with permafrost cannot be obtained here. The experiment can however give a bit of information on the associated change in pCO2 and D14C for a given terrestrial carbon release, but it is imperative to know the change in terrestrial carbon (in GtC) separated into vegetation and permafrost. There is no discussion on the reasons behind the possible changes in permafrost, their timing . . ... As such I don't really see the added value of the permafrost section. Also please see Kohler et al. 2014 (Nat. Comm.).

Thanks for pointing this out, such information is indeed lacking in the paper. We will do this by adding another figure in section 3.2 (Impacts of permafrost and biosphere) showing time series of vegetation carbon, permafrost carbon and their sum (total terrestrial carbon) across the ALL_TF experiment. This will then serve to extend the discussion on this topic and to evaluate this effect in our simulations with respect to literature. Thanks for your suggestions on literature.
An in-depth discussion on the reasons behind changes of permafrost and their timing, however, was not intended within this study and would go beyond the scope of the latter and the possibilities of the model. Despite the fact that it does

not add new value to permafrost research as such, the permafrost section is still well motivated due to its importance for the discussed processes and the model development work that had been conducted as part of the study.

5. References:

- The introduction is imprecise and lacks a lot of important references for the work described. A lot of work on glacial/interglacial changes in pCO2 is not mentioned (see work by F. Joos team for example). A lot of references on C14 workis missing (Kohler et al. 2014, Huiskamp and Meissner 2012).

  Thanks for the suggestions, we will include some more literature if applicable. Your statement (first sentence) is very general. We will revise the text after taking into account your comments, however, please specify if there are particular issues in the introduction.

- Some references are also used in the wrong context: For example p 18, the reference to Burke and Robinson (2012) is completely misused as they present D14C data from the Southern Ocean and not d13C data. As such the whole paragraph p18, L3-14 is wrong and please note that there is no d13CO2 dip during the B/A.

  Thank you for reading carefully, those are two slips of the pen. The reference should be Schmitt et al. (2012) here and as such, the paragraph makes sense again. And it is a short "rise" of $\delta^{13}C_{atm}$ during the B/A. We will correct that.

- P3, L9-10 in the intro is wrong.

  Thanks, it will be changed to:
  Okazaki et al. (2010) argue that North Pacific water masses could help to explain the release of old carbon during the last glacial termination and Kwon et al. (2012) point out the possible influence of deep North Atlantic water masses for this process.

- P6, L9 is wrong
  Thanks for reading carefully, it will be corrected to:
  Hence, iron fertilization probably led to enhanced new production of organic
  matter in the high latitude ocean during the LGM

---

## Author Comment (AC2) · 15 Mar 2016

**Reply to:**
*on "A model-data assessment of the role of Southern Ocean processes in the last glacial termination" by R. Eichinger et al. (cp-2015-190)*
**from Anonymous Referee #2**

[Figure]

*Dear Anonymous Referee #2,*

*we appreciate your valuable comments and suggestions. Please find our answers (in blue) to your comments (in black) below:*

This study uses a simplified numerical climate model to assess the role of different processes on the last glacial termination. This is an important topic for which many questions remain. Although it is a good idea to combine data and model to better understand the changes taking place at the termination, I am not sure the tool used here is suited for the task. The model, including new changes and shortcomings due to its simplicity, should be better explained. Other studies have focused on the last deglaciation with more complex models, this paper should better explain what is new here compared to previous work. It might need to be re-written in a clearer way.

General comments:

- Model used: I have serious concerns regarding the suitability of the numerical model used: it's a simplified model with only one hemisphere, a simple atmosphere (EBM) and no real ocean dynamics. The terrestrial biosphere model only depends on temperature (not precipitation). Is it enough to draw conclusions on changes impacting the carbon cycle on Earth based on changes in terrestrial biosphere and ocean dynamics?

  Please see below for the explanations to the specific points made here.

- Methods: Several changes have been made on the model. They seem important for the study and should not be in the supplementary material, but in the main text as they are relevant to the results.

  Thanks for reading carefully, but we only partly agree here. Some specific points are indeed missing in the main text and we will include those, however, for the

sake of readability of the paper would prefer to keep the rather technical parts in the Supplement. See below for details.

- Novelty: what is new in this study? The permafrost part has not been studied before, but this is not the main topic of the study (the Southern Ocean) and the permafrost module seems very simple and is not validated. The other mechanisms have already been studied in the past with better suited models, especially changes in ocean dynamics with models that better simulate the ocean dynamics (Tschumi et al., 2011; Bouttes et al., 2012; Brovkin etal., 2012; Menviel et al., 2012; Mariotti et al., 2016). The most interesting and new part is probably the section on carbon 14 and the role of the production rate, but then the paper should be re-organised around this, and the new work by Mariotti et al. (2016) discussed.

  Thanks, we will make more clear what the novelties of this study are, we also give a list of those below. The permafrost section will be extended (see also reply to ref #1).

- The main process that is studied is the change of mixing in the ocean. But the model-data comparison only focusses on atmospheric carbon isotopes. Since the main change comes from the ocean, it would be better to also compare model-data for carbon isotopes in the ocean.

  We believe that comparisons with atmospheric carbon isotopes are to be preferred since good data for these are available and since they represent global averages since the atmosphere is relatively well mixed on the time scales we are considering. But we take your point and now will also include more discussion in our Sect 2.1 of comparisons with ocean carbon isotope data.

Specific comments:

- In the abstract (and in other sections) the authors are very vague on the different processes and mechanisms studied such as p. 1 lines 4-5: "this interplay of processes" p.1 line 12: "various mechanisms" Could you be more precise?

  Thanks, this will be revised to be more precise..

- Can you also be more precise concerning the variables you're looking at? P.1 line 14: "the atmospheric variations": what are you referring to?

  We will include: ... in atmospheric temperatures, $pCO_2$ and carbon isotope ratios...

- p.1 line 21: "other [. . .] mechanisms": can you be more precise?
  p.1 line 21: "also contribute to the overall climate change": are you talking about the climate change or the changes in the carbon cycle? The same issue arises several times in the text, there seems to be a confusion between carbon cycle and climate.

  We will be more specific with something like: "This includes changes in ocean dynamics as well as in biogeochemical properties like variations in the phosphate inventory. Also the variability in atmospheric conditions, such as the dust concentration that has an impact on oceanic iron fertilisation and the radiative forcing, and terrestrial biosphere changes and permafrost contribute to the overall change in the carbon cycle and thus in the Earth's climate system. However, the individual contributions of these processes and their interactions remain unclear (Kohfeld and Ridgwell, 2009) and a comprehensive explanation for the atmospheric pCO2 rise across the last glacial termination is still lacking."

- p.3 line 13: The reference for LGM high salinity should be Adkins et al., 2002, the one for d13C: Curry and Oppo (2005), not Bouttes et al. (2011).

Thank you very much for pointing this out, will be corrected.

- p.3 line 22: We HAVE also DEVELOPED a set of functions. . .

  Thanks, will be corrected.

- p3 line 23: I'm not sure I agree with the fact that previous studies have looked at only one mechanism while this one would have a more comprehensive approach. I think the novelty of the study here compared to previous ones should be better explained, and specifically what is different here. Other models were already taking into account different mechanisms (ocean dynamics, biogeochemistry in the ocean, terrestrial biosphere. . .) with better suited models. The only thing that seems new to me is the inclusion of permafrost, but very few is said about it. Other similar work with models that should be discussed: Brovkin et al., 2012; Menviel et al., 2012; Mariotti et al., 2016. Rather than Bouttes et al. (2011), the comparison should be with the study on transient simulations (Bouttes et al., 2012).

  We will revise this sentence to read "...rather than concentrating on a subset of specific mechanisms... ." Furthermore, in our revision we will better describe what is new and/or different in our approach compared to the other studies. Among such novelties are 1. simultaneous calculation and presentation of global mean temperature, $pCO_2$, atmospheric and oceanic carbon isotopes and ocean dissolved oxygen concentration. 2. Continuous time series of all these properties across the "Mystery interval" and not just snapshots of before and after, 3. A comprehensive treatment of the role of dust and 4. inclusion of permafrost (see above).

- p. 4 line 4: Given that this study aims at studying the role of Southern Ocean processes, isn't it an issue that the model has only one hemisphere?

  This is a limitation of the DCESS Earth System model and as such it can not reproduce bipolar see-saw like effects. We do discuss this and this is one reason

why we concentrate on the "Mystery Interval" (with its evidence for a strengthening of Southern Ocean vertical exchange) and do not address subsequent seesaw-like variability (Bølling-Allerød; Younger Dryas), most likely due to changes in the strength of the Atlantic MOC. Furthermore we note that although the model only has one hemisphere, the high latitude ocean zone that it does have is not unlike the Southern Ocean with its wide latitudinal extent and the bounding of it by land poleward of 70 degrees.

- p. 4 line 6: Is the use of an EBM sufficient to correctly represent the terrestrial biosphere changes during the termination?

We feel that our new terrestrial biosphere module (TBM) is well dimensioned to fit in with the other simplified Earth System modules of the model. We do not calculate precipitation in our model but rather our TBM is based on an emulation of a complex terrestrial biosphere model (LPJ model) that includes the effects of precipitation. Our temperature dependence of latitudinal boundaries of vegetation zone thus includes implicitly the role of precipitation. In our supplement we provide a detailed evaluation of the performance of this new module and will mention this in more detail in the main text.

- p.4 line 8: what are the "anthropogenic activities" for the deglacial simulations? If it's not relevant for this study it should not be mentioned. Also how are the volcanism and weathering taken into account for this period?

"...and anthropogenic activities..." will be removed from this line. One of the features of the DCESS model is that it is an open system model that considers input from weathering and volcanism and outputs from burial. There is no clear consensus with regard to weathering changes since cooler LGM temperatures may favour less weathering while lower sea level and exposed shelves together with the action of the ice sheets may favour more weathering. Volcanism may have been affected by lowering sea level and ice sheet loading but this remains

somewhat speculative. Given this state of affairs we have chosen in our paper to maintain weathering and volcanism constant at the values found for the pre-industrial calibration of Shaffer et al. (2008).

• p.4 lines 16-27: Does the definition of the three additional zones have any impact on the carbon cycle (amount of carbon stored, isotopic fractionation) and /or on climate (albedo. . .) or is it just an output to compare with data? The new changes are presumably important for the study and might be what is new compared to others (especially the permafrost): the developments should be included in the text and not be put in the supplement.

Indeed, the zones have an impact on the carbon storage and thus on climate (different albedos have not yet been implemented for the sections, though.), see Supplement Sect. 3.3. We did miss to include that in the main text, thanks for pointing it out. We will include some new text on this in the revision.
Most of the description of the new (biosphere) developments are rather technical and hence would disturb the reading flow of the manuscript. Therefore, we prefer to keep those parts in the Supplement. However, we will add some more explanation on the parameterisation of permafrost in the main text.

• p.5 line 3: specify which proxy-records

We will be more specific here and mention some of the references that are discussed below in the section.

• p.5 lines 5-10: could you explain the physical rationale to have such a diffusion profile with a sharp reduction with depth (apart from changing the mixing), i.e. why would the mixing be different at the LGM? Given that there is only one hemisphere, is it a problem that you do it for the entire high latitude ocean and not only the Southern Ocean?

There are multiple lines of evidence to indicate isolated deep water during the LGM (e.g. Broecker and Barker, 2007, Burke and Robinson, 2012). In the

real Southern Ocean this may reflect some combination of increased stratification at some intermediate depth that would inhibit deep convection and weakening/shallowing of the deep upwelling. In the context of the DCESS model this can be addressed with our prescribed diffusion profile. Also see the comment above on the analogy of our high latitude ocean to the real Southern Ocean and our reply to referee #1.

- p.5 line 12: Given that the change of the diffusion profile is the main process studied, it should be explained here and not in the supplement. In the supplement you do not explain what variables were used to find the best guess profile. This should be shown (in the main manuscript) with the comparison with data.

  In the Supplement we do explain what variables we change to get the best-guess profile ($p_a$: transition depth). You are right that more should be explained in the main text as we will do in the revision. For the sake of readability of the main text, however, we would still prefer to keep the mathematical description of the profile in the Supplement (see also below).

- p. 6 lines 33-34: Explain the storage of carbon below the ice sheet and in permafrost and what data are used to constrain it.

  We will add some explanation on this in the main text too (also see above and our response to reviewer #1).

- p.7 lines 12-14: can you give the values from the data to compare with the model results?

  Thanks, we will do that and/or refer to table 2.

- Figure 2. Please add a,b,c. . . for each panel. Could you also add the legend (blue, red, black). The units should be given in brackets. As the goal is to compare with data, can you add the data for the variables for which they exist, such as the carbon isotopes?

a,b,c,... and legend will be added.
According to the Standard Institute, brackets around units are only needed if ambiguity has to be avoided (please see http://physics.nist.gov/cuu/rules.html)
We will try to extract some $^{13}$C and $^{14}$C data and include it in Fig. 2

- Table 2 and in other places in the text: for oceanic d13C data use Curry and Oppo (2005), Hesse et al. (2011), Peterson et al. (2014).

  Thanks, this will be done.

- Figure 3. Put units in brackets.

  See above.

- p.11 line 18: given that the change of diffusion is the main mechanism studied and most of the results come from its change, it should not be detailed in the supplement but in the main text.

  We feel that there is enough detail in the main text for understanding the concept of this mechanism, in particular through Fig. 1. It is only the detailed mathematical description that we banished to the Supplement. Hence, for the sake of readability of the main text, we would like to keep this as it is, however, we will revise and/or extend our explanations to avoid misunderstandings (see also reply to ref #1).

- Figure 4. add a,b,c for each panel and units in brackets.

  a,b,c,... will be added, units see above.

- p.13 line 2: can you give the changes of carbon stored in the terrestrial biosphere? Is this in line with data and previous simulations? (e.g. Ciais et al., 2012)

We will add another figure and some more discussion on this topic (see reply to reviewer #1). Some information on this is also already given in the Supplement. Thanks for the additional reference.

- Figure 5 and 6 . add a,b,c for each panel and units in brackets.

  a,b,c,.. will be added, units see above.

- p.19 lines 5-6: this is not true anymore, see Mariotti et al., 2016.

  Thanks, we will consider this very recent paper too.